# Reducing risk behaviours after stroke: An overview of reviews interrogating primary study data using the Theoretical Domains Framework

**Patricia Hall**[1,2]*, **Maggie Lawrence**[3], **Thilo Kroll**[4], **Catherine Blake**[1], **James Matthews**[1], **Olive Lennon**[1]

**1** School of Public Health, Physiotherapy and Sports Science, Health Science Centre, University College Dublin, Dublin, Ireland, **2** iPASTAR (Improving Pathways for Acute Stroke and Rehabilitation) Collaborative Doctoral Award, Division of Population Health Sciences, Royal College of Surgeons in Ireland, Dublin, Ireland, **3** Department of Nursing and Community Health, Glasgow Caledonian University, Glasgow, United Kingdom, **4** School of Nursing, Midwifery and Health Systems, Health Science Centre, University College Dublin, Dublin, Ireland

* patricia.hall@ucdconnect.ie

**Data Availability Statement:** All relevant data are within the manuscript and its Supporting information files.

## Abstract

### Background

Lifestyle changes, in addition to preventive medications, optimise stroke secondary prevention. Evidence from systematic reviews support behaviour-change interventions post-stroke to address lifestyle-related risk. However, understanding of the theory-driven mediators that affect behaviour-change post-stroke is lacking.

### Methods

Electronic databases MEDLINE, Embase, Epistemonikos and Cochrane Library of Systematic Reviews were searched to March 2023 for systematic reviews addressing behaviour-change after stroke. Primary studies from identified systematic reviews were interrogated for evidence supporting theoretically-grounded interventions. Data were synthesized in new meta-analyses examining behaviour-change domains of the Theoretical Domains Framework (TDF) and secondary prevention outcomes.

### Results

From 71 identified SRs, 246 primary studies were screened. Only 19 trials (N = 2530 participants) were identified that employed theoretically-grounded interventions and measured associated mediators for behaviour-change. Identified mediators mapped to 5 of 14 possible TDF domains. Trial follow-up ranged between 1–12 months and no studies addressed primary outcomes of recurrent stroke or cardiovascular mortality and/or morbidity. Lifestyle interventions targeting mediators mapped to the TDF *Knowledge* domain may improve the likelihood of medication adherence (OR 6.08 [2.79, 13.26], $I^2$ = 0%); physical activity participation (OR 2.97 [1.73, 5.12], $I^2$ = 0%) and smoking cessation (OR 10.37 [3.22, 33.39], $I^2$ =

**Funding:** Health Research Board, Ireland, https://www.hrb.ie/ Collaborative Doctoral Award iPASTAR (improving Pathways for Acute Stroke and Rehabilitation) [CDA-2019-004]. The first author (PH) is a PhD scholar funded under this program. Funders did not play any role in the study design, data collection and analysis, decision to publish, or preparation of the manuscript.

**Competing interests:** The authors have declared that no competing interests exist.

20%) post-stroke, supported by low certainty evidence; Lifestyle interventions targeting mediators mapping to both TDF domains of *Knowledge* and *Beliefs about Consequences* may improve medication adherence post-stroke (SMD 0.36 [0.07, 0.64], $I^2$ = 13%, very low certainty evidence); Lifestyle interventions targeting mediators mapped to *Beliefs about Capabilities* and *Emotions* domains may modulate low mood post-stroke (SMD -0.70 [-1.28, -0.12], $I^2$ = 81%, low certainty evidence).

## Conclusion

Limited theory-based research and use of behaviour-change mediators exists within stroke secondary prevention trials. *Knowledge*, *Beliefs about Consequences*, and *Emotions* are the domains which positively influence risk-reducing behaviours post-stroke. Behaviour-change interventions should include these evidence-based constructs known to be effective. Future trials should address cardiovascular outcomes and ensure adequate follow-up time.

## Introduction

Stroke, the leading cause of adult acquired disability, has a 20% risk of recurrence within five years [1]. A quarter of these recurrent events occur in the first year post-stroke, with an average annual risk thereafter of 4% [2]. Poorer prognosis and greater disability are associated with recurrent events, highlighting the need for effective risk reduction measures [1, 3]. Population attributable, modifiable risk factors account for up to 90% of stroke risk [4]. These include: hypertension, physical activity, dyslipidaemia, diet, central adiposity, psychosocial factors (home and/or work stress, life events and depression), current smoking, cardiac causes, high/heavy episodic alcohol consumption and diabetes mellitus. Modifiable risk factors for (recurrent)stroke are amenable to lifestyle change [5]. A modelling study suggests that lifestyle changes in diet and physical activity levels alongside optimized pharmacotherapy, could reduce five-year recurrent event rates after stroke by 80% [6]. Initiating and sustaining lifestyle changes to mitigate the risk of recurrent events constitute complex behaviours entailing multiple interacting components at individual, social and environmental levels [3, 7, 8]. Aspects such as knowledge, intentions, self-efficacy, motivation, outcome expectancies, perceived susceptibility/severity and social influences can act as barriers or facilitators to effective change [9]. These are classified as determinants of behaviour and can act as mediators for change, defined as 'the intermediary variables in the causal process between an intervention and the behaviour change effect' [10].

Whilst published SRs broadly address the efficacy of lifestyle-based interventions post-stroke [11–15], uncertainty remains about what works and why. Inclusion of studies lacking theory and understanding of determinants of health-related behaviours in stroke, may have influenced their findings. Failing to critically examine the role of these determinants as the mediators to affect behaviour, limits our understanding of effective interventions in stroke secondary prevention. An aligned overview of reviews [16], providing a best-evidence synthesis of behaviour-change interventions after stroke, highlighted the further need to delineate theory-based interventions and provide greater insight into why some interventions identified were effective or ineffective. This requires a better understanding of the role of theory and associated mediators in affecting behaviour-change [17, 18]. International secondary prevention guidelines recommend using theory-based lifestyle interventions [19], mirrored in the Medical

Research Council (MRC) guidance on complex interventions where program theory is deemed essential to maximize the efficiency, use and impact of behaviour-change research [20].

Explicit use of theory in the design and evaluation of interventions in stroke secondary prevention presents this opportunity to understand why interventions work, for whom, and in what context [18]. Selecting one or more theories as the basis for intervention development can prove challenging, partly due to often overlapping theoretical constructs [21]. The Theoretical Domains Framework (TDF) is a comprehensive theory-informed approach to understanding the determinants of behaviour change and the factors influencing intervention development. The TDF was developed to allow theories and their constructs to be synthesised into groupings to make behaviour-change theories more accessible in intervention design and analysis [21, 22]. This is important as it provides a systematic and rigorous framework for understanding behaviour change in multiple populations and settings. Comprising 87 component parts across fourteen overarching domains of *Knowledge*, *Skills*, *Social/Professional Role and Identity*, *Beliefs about Capabilities*, *Optimism*, *Beliefs about Consequences*, *Reinforcement*, *Intentions*, *Goals*, *Memory, Attention and Decision Processes*, *Environmental Context and Resources*, *Social Influences*, *Emotions*, and *Behavioural Regulation* [22], the TDF provides comprehensive coverage of the possible mediators influencing behaviour-change. Tabular representation of the TDF describes these domains as they pertain to stroke secondary prevention in the current study (S1 Table).

This overview of reviews aims to unpack complex interventions identified in primary studies across published SRs addressing behaviour-change post-stroke, to enable a better understanding of the underlying factors necessary to achieve the desired outcomes. Included primary studies are interrogated to identify the role of theory and proposed mediators for change, as mapped to the TDF domains. New meta-analyses synthesize primary studies by behaviour-change theoretical domains and secondary prevention outcomes, allowing the certainty of existing evidence supporting the working components of interventions to be examined. To our knowledge, the TDF has not previously been used as the basis from which to understand the behaviour-change constructs of effective stroke secondary prevention interventions.

## Specific objectives

- Provide a compendium of theoretically-grounded primary studies identified across published SRs addressing secondary stroke prevention, using behavioural/self-management interventions.

- Extract the active components of reported interventions under the Template for Intervention description and Replication (TIDieR) [23] checklist, notably the theoretical perspectives described and the mediator/s for behaviour-change identified and measured.

- Synthesize the results from the identified primary studies by behaviour-change theoretical domains and secondary prevention outcomes.

- Determine the quality of evidence supporting the behaviour-change interventions and their working components by assigning a Grading of Recommendations Assessment, Development and Evaluation (GRADE) [24] of evidence for meta-analyses conducted.

## Methods

This overview of reviews adheres to the preferred reporting guideline for overviews of reviews of healthcare interventions (PRIOR) statement and checklist (S2 Table) [25]. It was preceded

by an *a priori* published protocol [26] which detailed a two-phased approach. Phase one, published elsewhere [16], identifies, synthesises and provides GRADE [24] of certainty for published meta-analytic evidence. Phase 2, reported here, first identifies primary study level data from all published SRs employing theoretically-grounded behaviour-change and/or self-management interventions and then synthesises these data in new meta-analyses to examine evidence supporting the mediators for behaviour-change in affecting positive changes in secondary prevention outcomes. A theoretically-grounded study was considered, for the purposes of this review, to be one that identified a theoretical perspective and identified and measured mediator/s for the targeted behaviour-change [27] that mapped to the TDF domains.

## Inclusion/Exclusion criteria

SRs of randomized control trials (RCTs) or cluster RCTs (CRCT) testing interventions for behaviour-change and/or self-management of risk in stroke secondary prevention were first identified. Primary studies included in these reviews were then considered where the following were detailed:

- Adult population comprising stroke/TIA

- Intervention/s targeting stroke risk reduction at an individual or population level

- Intervention/s identifying a theoretical perspective and measuring a stated mediator for behaviour-change that mapped to the TDF

- Comparators of usual care, placebo, sham, or other intervention

- Outcomes recorded that addressed mortality, recurrent stroke or other cardiovascular events, or secondary outcomes addressing any one or combination of the following health behaviours–secondary prevention medication adherence, healthy diet, physical activity participation, smoking cessation, safe alcohol consumption and emotional self-regulation.

 Exclusion criteria applied:

- Interventions designed to alter care process or health professionals' education/practice.

- Interventions not targeting behaviour-change in stroke secondary prevention.

- Telehealth interventions

- Interventions targeting family/partner dyads, unless behaviour-change in the person with stroke was specifically targeted and extractable.

## Search strategy

Using a comprehensive search strategy compiled in conjunction with a liaison librarian, electronic databases MEDLINE, Embase, Epistemonikos and Cochrane Library of Systematic Reviews were systematically searched from inception to March 2023 with no limitations applied. For databases not specific to systematic reviews (Medline, Embase), a third methodological search string for systematic reviews was included. These two databases were chosen as they are two of the largest health focussed databases and we were confident, based on our experience and previous searches that they would contain the reviews we were looking for. In addition, reference lists of included SRs were checked. It is possible that more recent RCTs, not yet reviewed in SRs are not included. The full search strategy which targeted published systematic reviews is provided (S1 File).

## Screening and selection

Identified SRs were screened at title and abstract stages by two independent reviewers. An inclusive approach was taken whereby if it was unclear whether the SR met the inclusion criteria, it progressed to the next stage. Full manuscripts were next reviewed independently for inclusion by two reviewers. SRs were included where they reported RCTs of behaviour-change/self-management interventions and reported secondary prevention outcomes of mortality, recurrent stroke, other cardiovascular events; health/lifestyle behaviours; emotional self-regulation; as summarized in the stroke secondary prevention model underpinning the overview (S1 Fig).

While phase 1 of this overview [16] excluded SRs that did not meet explicit criteria (i.e. no meta-analyses or where meta-analyses included mixed populations), phase 2 included any SR that contained primary RCTs of interest. All primary studies were subsequently screened for eligibility by two reviewers (PH, OL) to ensure they met the specific RCT study-level inclusion criteria listed above and that none of the following exclusion criteria applied:

- Interventions designed to alter care process or health professionals' education/practice.

- Interventions not targeting behaviour-change in stroke secondary prevention.

- Telehealth interventions

- Interventions targeting family/partner dyads, unless behaviour-change in the person with stroke was specifically targeted and extractable.

Following screening of RCTs from identified SRs, each trial intervention, as described in the paper, was examined for inclusion against domains 1 and 2 of the TIDieR checklist [23]. These relate specifically to providing the rationale for the intervention and the theoretical perspective. Where theory and rationale were provided, the next level screening identified whether a mediator for behaviour-change, mapping to the TDF, was identified and measured. Studies that did not measure proposed mediator/s pre and post intervention were excluded. Any disagreements regarding study eligibility were resolved by discussion to reach consensus.

## Data extraction

Data from included RCTs were independently extracted and cross-checked by two reviewers (PH, OL). These data included TIDieR checklist domains (1–10); mediators for behaviour-change; TDF domain of the mediator; participants' characteristics (e.g. age, gender, time post-stroke); reported stroke secondary prevention outcomes.

## Quality appraisal

The Cochrane Risk of Bias (ROB) [28] tool was used to assess the methodological quality of included RCTs. Where trials were appraised using this tool in the SR of origin, it was accepted and extracted by reviewers. Where more than one SR presented ROB for the same trial and a discrepancy between reviews was identified, the more conservative ROB was recorded. Where trials were appraised in the SR of origin using an alternative tool or where no appraisal was documented, two reviewers (PH, OL) independently appraised the trial using the Cochrane ROB tool to ensure conformity.

## Data synthesis

Meta-analyses, conducted using Review Manager 5 (RevMan5) [29], grouped data by TDF domains and secondary prevention outcomes, where data presented permitted. For

continuous data, where different scales assessed the same outcome, standardized mean differences (SMD) with 95% confidence intervals (CI) were calculated. SMD is used as a summary statistic to measure effect size that quantifies differences in standard deviations between two groups. The Hedges' g version of SMD conducted here in RevMan5 is the preferred statistic when sample sizes are unequal and/or are small ($< 20$), as the case in the current study, as it takes each sample size into consideration when calculating the overall effect size. The inverse variance method was used as it is especially suitable when using SMD to minimise uncertainty of the overall effect size [30].

For dichotomous variables, odds ratios (OR) with 95% CIs were employed using the Mantel-Haenszel method. Random effects models were applied to provide a more conservative estimate of overall effect size as statistical heterogeneity was assumed [30]. The $I^2$ statistic measured heterogeneity; >50% was considered substantial [30]. To over-come a unit-of-analysis error, where multiple comparisons from the same study were included more than once in a meta-analysis, the group was split into separate groups with smaller sample sizes to avoid over-counting [31]. Where included trials reported outcomes measured at more than one time point post-intervention, the last follow-up time was included in the meta-analysis. Where possible, sensitivity analysis was conducted, to examine pooled secondary prevention outcome data only from trials where the mediator for behaviour-change was observed to positively change. The certainty of the evidence for each effective intervention/outcome group identified was evaluated using the GRADE criteria [24] and agreed by consensus of two reviewers (PH and OL).

## Results

### Study identification and selection

The PRISMA [32] flow diagram (Fig 1) details the overview flow process, including reasons for SR and subsequent primary RCT exclusion. As detailed, 71 reviews were included, yielding 246 unique RCTs. Of these, nineteen met the full criteria for inclusion [33–52]. Two papers which reported the same study at 3 and 12 months [49, 50] were considered as one trial. One study could not be retrieved, despite inter-library agency [53].

### Description of included primary RCTs

Table 1 summarises the characteristics of all nineteen included RCTs. The trials were conducted across four geographical locations–Australia [33, 38, 40], North America [34, 36, 48], Asia [44, 45, 51, 52] and Europe [35, 37, 39, 41–43, 46, 47, 50]. When broken down by country and world bank classification all but two studies from an upper-middle-income economy (China) [51, 52] originated in high-income nations [54]. A total of 2530 participants with confirmed first or recurrent stroke/TIA were included across the trials. Seventeen trials included men and women, one included men only [41], and one did not report the gender of participants [35]. Usual care was the comparator in most trials; however, four trials compared the intervention to alternate/active controls (e.g. computerised cognitive training [41]; education program [47]; stress management program without mindfulness [51]; non-medication related conversation [43]. One trial was a three-arm trial [38] but only data from cognitive-behavioural therapy versus usual care is included here based on outcomes reported. Intervention follow-up ranged from immediate post-intervention to twelve months across the included RCTs.

The interventions, as described in each study, differed considerably in terms of their rationale, active components, delivery, and theoretical underpinning, as documented under all TIDieR domains (S3 Table). Secondary prevention outcomes that mapped to the prevention

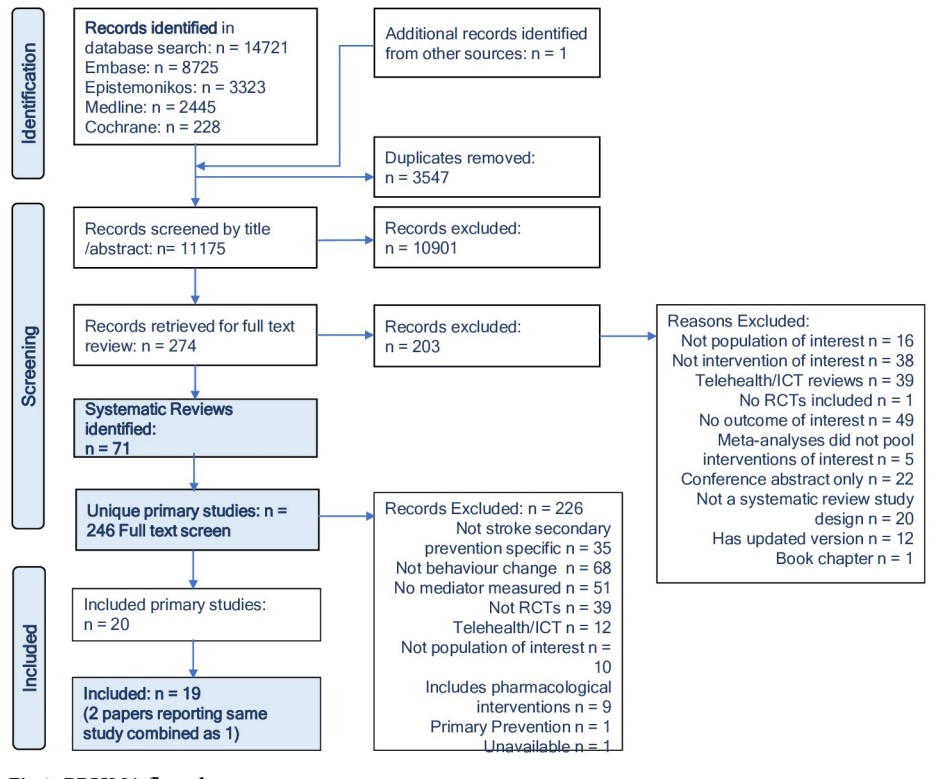

**Fig 1. PRISMA flow chart.**

model (S1 Fig) [26] were extracted. Twelve stroke secondary prevention outcomes drawn from seven trials documented statistically significant effects. Significant heterogeneity of measures and scales employed across trials was evident, limiting meta-analyses that could be conducted.

Across the nineteen RCTs included, eight different mediators for behaviour-change were referenced and measured pre and post intervention. These mediators mapped to five of fourteen possible domains of the TDF, depicted in Fig 2. Eight trials [34, 42–46, 51, 52] demonstrated a positive change in the mediator for change between the intervention group and control, of which five reported statistically significant effects in outcomes related to stroke secondary prevention [34, 43, 45, 46, 52]. Others demonstrated improvement in both trial arms [36, 41].

## Quality appraisal

Risk of bias summary and graphical illustration of ROB assessments presented as percentages are included in supplemental material (S2 and S3 Figs). Principal sources of bias related to blinding of participants and personnel delivering interventions. As it is generally accepted that blinding of participants or interventionists would not be feasible in complex interventions, this criterion was not judged in the overall rating. Low ROB was judged where the criterion was met in remaining domains [46, 48]; an unclear or moderate risk was judged where there was low or unclear ROB for these domains [36, 37, 41, 43]; a high ROB was judged where there was high ROB in one or more domain [33–35, 38–40, 42, 44, 45, 47, 50–52].

Table 1. Study characteristics and findings.

| Study | Participants | Intervention theoretical perspectives | Brief description | Mediator | TDF Domain | Intervention | Time to follow-up | Outcomes measured | Key findings |
|---|---|---|---|---|---|---|---|---|---|
| Eames [33] 2013 Australia | N = 77 I: n = 37, 55% female Age: 55.2 (SD 16.7) C: n = 40, 54% female Age: 61.4 (SD 12.7) | Health Belief Model | Tailored stroke education and support package | Stroke knowledge No difference | Knowledge | Education and support package in addition to usual care; Pre-discharge face-to-face session; online information booklet; verbal reinforcement | 3 month follow-up | Mood (HADS) Self-efficacy | Significantly better self-efficacy for accessing information |
| Evans-Hudnall [34] 2014 USA | N = 52 I: n = 27, 42% female Age: 56 (SD 9.9) C: n = 25, 35% female Age: 46.6 (SD 10.7) | Cognitive behaviour therapy (CBT) to facilitate behaviour change | Tailored information and goal-setting self-care among underserved ethnic minority individuals | Stroke knowledge **Positive change in intervention group** | Knowledge | Secondary stroke prevention self-care education; Pre-discharge face-to-face; Post-discharge telephone follow-up | 1 month follow-up | Mortality Medication adherence (no's compliant) Fruit & vegetable consumption Exercise (minutes) Tobacco & alcohol use | Significantly reduced tobacco use and improved alcohol use |
| Gillham [35] 2010 UK | N = 52 I: n = 26 Age: 67.7 (SD 12.0) C: n = 26 Age: 68.9 (SD13.2) | Transtheoretical Model Stages of Change with Motivational Interviewing techniques | Enhanced secondary prevention education targeting readiness to change behaviour | Readiness to Change No difference | Intentions | Enhanced individual stroke risk factor education; Initial post-stroke clinic interview using motivational interviewing techniques and telephone follow-up | 3 month follow-up | Mood (HADS) Fruit & vegetable consumption Exercise (frequency) Alcohol use | Significant change is self-reported diet and exercise behaviour |
| Green [36] 2007 Canada | N = 52 I: n = 97, 42% female Age: 66.3 (SD 12.4) C: n = 100, 41% female Age: 67.2 (SD 12.4) | Transtheoretical Model Stages of Change with Motivational Interviewing techniques | Educational counselling to increase stroke knowledge | Readiness to Change Shift from passive to active stage of change for both groups. | Intentions | One-to-one counselling on personal risk factors at initial post-stroke clinic; Lifestyle class within 2 months | 3 month follow-up | Mortality Stroke knowledge | Significant improvement in stroke knowledge |

(*Continued*)

**Table 1.** (Continued)

| Study | Participants | Intervention theoretical perspectives | Brief description | Mediator | TDF Domain | Intervention | Time to follow-up | Outcomes measured | Key findings |
|---|---|---|---|---|---|---|---|---|---|
| **Hjelle** [37] 2019 Norway | N = 322 I: n = 166, 40% female Age: 66 (12.1) C: n = 156, 42% female Age: 65 (SD 13.3) | Theory of Salutogenesis, Self-determination, Narrative Theories | Dialogue-based intervention to enhance psychosocial well-being | Sense of Coherence No difference | Emotions | Dialogue-based individual sessions in participants' home; Commenced within 1 month of acute stroke; Guided self-determination to empower decisions on psychosocial wellbeing | 6 month follow-up | Mortality Normal Mood (GHQ) Depression (Yale) | Psychosocial wellbeing improved in both groups. No significant benefit found |
| **Hoffman (a)** [38] 2015 Australia | N = 33 I: n = 11, 36% female Age: 63.6 (SD 13.0) C: n = 10, 40% female Age: 57 (SD 14.2) | Self-efficacy and Motivational Interviewing | Coping skills intervention to improve self-awareness and coping skills | Self-efficacy No difference | Beliefs about Capabilities | Cognitive behavioural coping skills approach; face-to face sessions commenced pre-discharge and in participants' home | 3 month follow-up | Mood (HADS) Self-efficacy Stroke knowledge | No clear influence on anxiety and depression symptoms detected |
| **Jones** [39] 2016 UK | N = 78 I: n = 40, 50% female Age: 61.8 (SD 16.0) C: n = 38, 34% female Age: 68.8 (SD 10.3) | Social cognition theory and self-efficacy principles | Integrated stroke self-management programme | Self-efficacy No difference | Beliefs about Capabilities | Bridges Community Stroke Self-management; Face-to-face home visits; Workbook | 3 month follow-up | Mood (HADS) Stroke self-efficacy | No significant differences Self-efficacy showed most sensitivity to change in intervention group |
| **Kendall** [40] 2007 Australia | N = 100 I: n = 58, 29% female Age: 66.4 (SD 10.9) C: n = 42, 38% female Age: 66.3 (SD 10.4) | Stanford Model of Chronic Disease Self-management | Psychosocial skill expansion using self-management education approach | Self-efficacy No difference | Beliefs about Capabilities | Community chronic disease self-management programme; Additional stroke specific sessions; Face-to-face community groups | 3, 6, 6, 12 month follow-up | Mood (stroke specific quality of life scale) Self-efficacy | No influence detected |
| **Kootker** [41] 2017 Netherlands | N = 61 2 interventions (active control), all male, median age 61 years I: n = 31 AC: n = 30 | Cognitive behaviour therapy (CBT) to facilitate change of irrational and negative thoughts | Individually tailored cognitive behavioural therapy to reduce depressive symptoms | Proactive coping competence No between group difference. Significant change in both groups | Emotions | CBT augmented with occupational /movement therapy versus Computerised cognitive training | 4, 8 month follow-up | Mood (HADS) Coping competence | Significant and persistent improvement in anxiety & depression for both interventions. CBT not superior |

(*Continued*)

**Table 1.** (Continued)

| Study | Participants | Intervention theoretical perspectives | Brief description | Mediator | TDF Domain | Intervention | Time to follow-up | Outcomes measured | Key findings |
|---|---|---|---|---|---|---|---|---|---|
| **McKenna** [42] 2015 NI, UK | N = 25 I: n = 11, 36% female Age: 62 (SD 13.5) C: n = 13, 54% female Age: 67.3 (SD 10.6) | Self-efficacy principles | Stroke self-management programme | Self-efficacy **Positive change in intervention group** | Beliefs about Capabilities | Bridges Community Stroke Self-management; One-to-one sessions promoting specific behaviours; Patient held workbook | 3 month follow-up | Mood (GHQ) Self-efficacy Stroke self-efficacy | Less decline in mood in intervention group |
| **O'Carroll** [43] 2013 UK | N = 58 with active control I: n = 29, 31% female Age: 68.4 (SD 11.3) AC: n = 29, 41% female Age: 70.7 (SD 10.5) | Self-regulation theory; Implementations intention approach | Brief intervention to increase medication adherence | Beliefs about medications **Positive change in intervention group** | Beliefs about Consequences | Cognitive/ educational & behavioural brief intervention; Face-to-face at home; Control received visits and non-medication related conversations | 3 month follow-up | Medication adherence (MARS: MEMS) Blood pressure Beliefs about medications Brief illness perception questionnaire | Significantly higher adherence to medication, correct dose, taken on schedule |
| **Sit** [45] 2007 China | N = 190 I: n = 107, 50% female Age: 62.8 (SD 10.2) C: n = 83, 37.5% female Age: 64 (SD 12.0) | Interactive learn-practise-feedback-learn approach | Community-based interactive stroke prevention education programme | Stroke knowledge **Positive change in intervention group** | Knowledge | Community based group interactive education sessions to empower self-care | 3 month follow-up | Medication adherence Salty food consumption Exercise participation (type & frequency) Stroke knowledge | Significant positive changes in medication adherence, dietary habits, physical activity participation |
| **Sit** [44] 2016 China | N = 210 I: n = 105, 47.6% female Age: 67.8 (SD 14.2) C: n = 105, 47.6% female Age: 70.7 (SD 13.9) | Theory of health empowerment; self-efficacy; self-management | Stroke self-management programme to enhance patients knowledge and skills | Self-efficacy **Positive change in intervention group** | Beliefs about Capabilities | Health empowered stroke self-management programme in parallel with rehabilitation | 6 month follow-up | Mortality Medication adherence Self-BP monitoring Self-efficacy | No significant differences in medication adherence |
| **Slark** [46] 2013 UK | N = 96 I: n = 47, 36% female Age: 65 (SD 12.1) C: n = 47, 47% female Age: 66 (SD 12.7) | MRC approach to complex interventions as a framework. Complex intervention designed to increase risk awareness and knowledge | Enhanced individualised risk awareness intervention | Stroke knowledge **Positive change in intervention group** | Knowledge | Individual risk awareness session with tailored information; Face-to-face as inpatient | 3 month follow-up | Medication adherence Diet Physical activity Tobacco use Alcohol use Physiological factors Stroke knowledge | Significant increased risk awareness and self-reported lifestyle changes |

(*Continued*)

**Table 1.** (Continued)

| Study | Participants | Intervention theoretical perspectives | Brief description | Mediator | TDF Domain | Intervention | Time to follow-up | Outcomes measured | Key findings |
|---|---|---|---|---|---|---|---|---|---|
| **Tielemans** [47] 2015 Netherlands | N = 113 2 interventions as active control I: n = 58, 55% female Age: 55.2 (SD 8.9) AC: n = 55, 40% female Age: 58.8 (SD 8.7) | Self-management based on proactive coping and action planning versus General Stroke Education programme | Stroke specific self-management project teaching pro-active coping | Proactive coping competence No difference. | Emotions | Stroke specific self-management programme; teaching proactive coping skills & action planning; Face-to-face in small groups | 9 month | Mood (HADS) Coping competence | Did not favour self-management over education intervention |
| **Towfighi** [48] 2020 USA | N = 100 I: n = 49, 41% female Age: 60 (SD 7) C: n = 51, 35% female Age: 57 (SD 10) | Healthy Eating and Lifestyle After Stroke (HEALS) conceptual model based on chronic disease self-management | Enhanced self-management programme to improve diet and physical activity | Readiness to Change No difference | Intentions | Lifestyle management programme; Health behaviours, risk factors, motivation addressed | 6 month follow-up | Fruit & vegetable consumption Tobacco use Physiological factors Readiness to change | No significant changes in outcomes, small effect sizes, a longer duration recommended |
| **Wang** [51] 2020 China | N = 134, 54% female Age: 59.9 (SD 10.6) 2 interventions as active control I: n = 67 C: n = 67 | Mindfulness-based cognitive therapy versus Stress management/no mindfulness | Mindfulness-based cognitive therapy to improve quality of life and poststroke depression | Trait mindfulness **Positive change in intervention group** | Emotions | 2-hour group sessions of Mindfulness-based cognitive therapy over 8 consecutive weeks | End of intervention | Mood (CES-D) Trait mindfulness (MAAS) | Positive effects on patients' depression, social well-being, and emotional well-being. |
| **Watkins** [49, 50] 2007, 2011 UK | N = 411 I: n = 204, 42% female Age: median 70 years C: n = 207, 41% female Age: median 70 years | Motivational Interviewing (MI) | Motivational interviewing to support and build patients' motivation to adjust and adapt after stroke | Beliefs and expectations of recovery No difference | Beliefs about Consequences | early post-stroke MI sessions to improve mood; Individual patient-centred counselling | 3, 12 month follow-up | Mortality Mood (GHQ) Beliefs and expectations | Improved mood and reduced mortality |
| **Zhang** [52] 2015 China | N = 89 I: n = 45, 35% female Age: 58.7 (SD 9.2) C: n = 44, 38% female Age: 59.3 (SD 8.1) | Mindfulness-based cognitive therapy | Mindfulness-based behavioural training to improve poststroke depression | Trait mindfulness **Positive change in intervention group** | Emotions | Mindfulness-based cognitive behavioural training; Intensive consolidation training | End of intervention | Mood (HAM-D) Trait mindfulness (MAAS) | Significantly decreased depression scores |

I: Intervention; C: Control; TDF: Theoretical Domains Framework; BCTs: Behaviour change techniques; HADS: Hospital anxiety and depression scale; GHQ: General health questionnaire; CES-D: Centre for Epidemiological Studies-Depression; HAM-D: Hamilton depression rating scale; MARS: Medication adherence rating scale; MEMS: Medication event monitoring system; MAAS: Mindful attention awareness scale; MI: Motivational interviewing; MRC: Medical Research Council

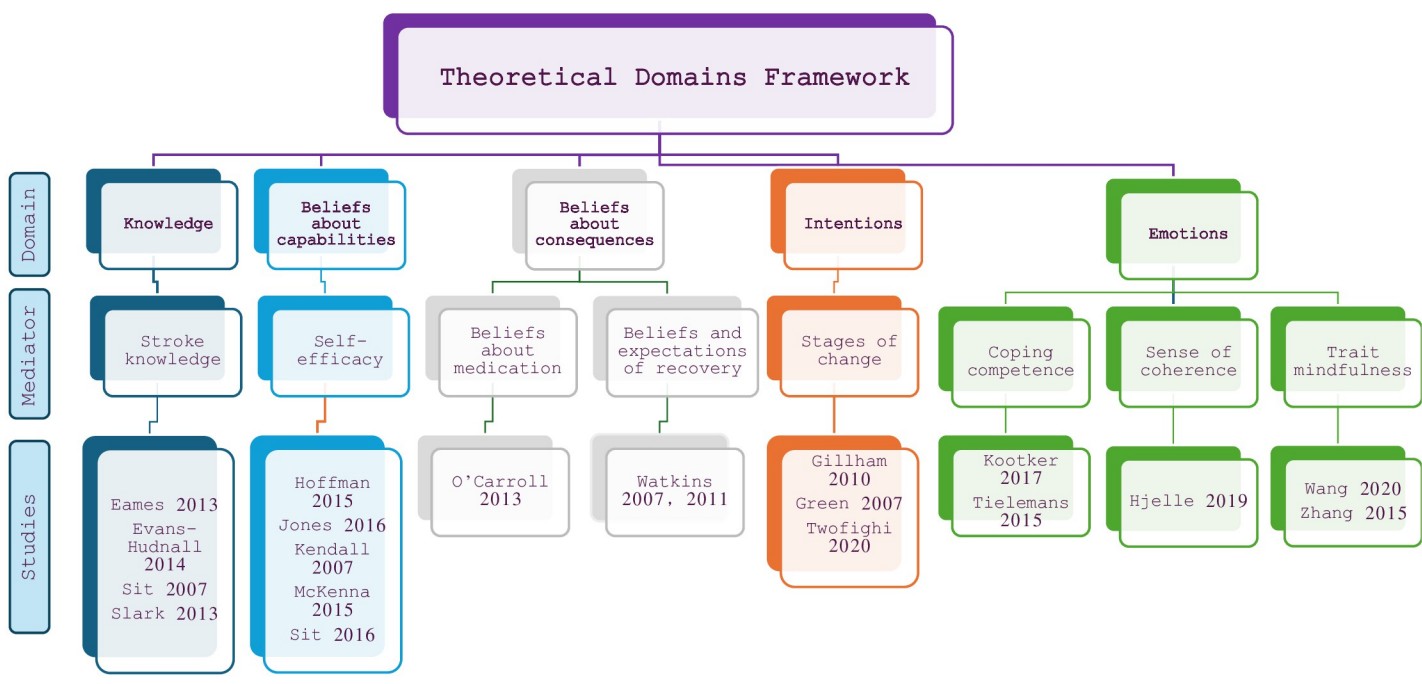

**Fig 2. Theoretical Domains Framework, mediators, and included studies.**

## Meta-analysis: Primary outcomes

**Mortality, recurrent stroke, cardiovascular events.** Despite describing their interventions as designed to reduce risk, no included trial reported outcomes of cardiovascular death, recurrent stroke or cardiovascular events, and follow-up was short ranging between 1–12 months. Mortality data, available from five trials [34, 36, 37, 45, 50], was reported under adverse events. Pooled data on 1203 participants (S4 Fig), as grouped by TDF domain, demonstrated no significant difference in the odds of death as an adverse event in lifestyle-related behaviour-change interventions when compared to controls (OR 0.60, 95% CI 0.32–1.11, p = 0.62, $I^2$ 0%).

## Meta-analysis: Secondary outcomes

Secondary outcomes addressed measures of lifestyle behaviour-change and emotional self-regulation. Seven trials [34, 35, 43–46, 48] reported outcomes of one or a combination of health behaviours addressing medication adherence, healthy eating, physical activity participation, tobacco and alcohol use. Twelve trials [23, 33, 35, 37, 39–42, 47, 50–52] reported emotional self-regulation outcomes utilizing measures of anxiety, depression or psychological/psychosocial distress. Table 2 provides the GRADE of evidence for all meta-analyses supporting stroke secondary prevention outcomes by theoretical domains.

## Health behaviour outcomes

**Medication adherence.** Five trials reported medication adherence as a risk reducing behavioural outcome [34, 43–46]. Data were pooled from two trials measuring frequency of self-reported adherence [34, 46]. Both trials used behaviour-change mediators mapped to the TDF *Knowledge* domain. Meta-analysis demonstrated a significant effect in favour of the

**Table 2. GRADE of evidence: Secondary prevention outcomes by theoretical domains.**

| № of studies | Study design | Risk of bias | Inconsistency | Indirectness | Imprecision | Other considerations | behaviour change interventions | usual care and/or active control | Relative (95% CI) | Absolute (95% CI) | Certainty |
|---|---|---|---|---|---|---|---|---|---|---|---|
| **Certainty assessment** | | | | | | | **№ of patients** | | **Effect** | | **Certainty** |
| **Medication adherence/*Knowledge* domain—(frequency of adherence)** | | | | | | | | | | | |
| 2 | randomised trials | serious[d] | not serious | not serious | serious[b] | none | 58/74 (78.4%) | 29/72 (40.3%) | **OR 6.08** (2.79 to 13.26) | **401 more per 1,000** (from 250 more to 497 more) | ⊕⊕◯◯ Low |
| **Medication adherence/*Knowledge* and *Beliefs about consequences* domains (assessed with: MARS & MMAS)** | | | | | | | | | | | |
| 2 | randomised trials | very serious[a] | not serious | not serious | serious[b] | none | 136 | 112 | - | SMD **0.36 higher** (0.07 higher to 0.64 higher) | ⊕◯◯◯ Very low |
| **Physical activity participation/*Knowledge* domain—achieving targets** | | | | | | | | | | | |
| 2 | randomised trials | serious[a] | not serious | not serious | serious[b] | none | 98/154 (63.6%) | 51/130 (39.2%) | **OR 2.97** (1.73 to 5.12) | **26 more per 100** (from 14 more to 38 more) | ⊕⊕◯◯ Low |
| **Smoking cessation/*Knowledge* domain** | | | | | | | | | | | |
| 2 | randomised trials | serious[c] | not serious | not serious | serious[b] | none | 31/74 (41.9%) | 7/72 (9.7%) | **OR 10.37** (3.22 to 33.39) | **43 more per 100** (from 16 more to 69 more) | ⊕⊕◯◯ Low |
| **Depression/*Beliefs about capabilities* and *Emotions* domains(assessed with: HADS, HAMD, MAAS)** | | | | | | | | | | | |
| 4 | randomised trials | very serious[a] | not serious | not serious | not serious | none | 159 | 151 | - | SMD **0.7 SD lower** (1.28 lower to 0.12 lower) | ⊕⊕◯◯ Low |

**CI:** confidence interval; **OR:** odds ratio; **SMD:** standardised mean difference

**MMAS:** Moriskey Medication Adherence Scale; **MARS:** Medication Adherence Rating Scale; **HADS:** Hospital Anxiety and Depression Scale; **HAMD:** Hamilton depression rating scale; **MAAS:** Mindful Attention Awareness Scale

Explanations:

[a.] high risk of bias in multiple domains assessed in Cochrane risk of bias tool

[b.] small number of included studies

[c.] studies were rated as unclear in multiple risk of bias domains and high risk of bias related to blinding

[d.] some concerns related to blinding

[e.] inconsistent effect direction across studies

intervention group (OR 6.08 [2.79, 13.26], p< 0.00001, $I^2$ 0%) (Fig 3a). GRADE criteria identified low certainty of evidence (Table 2). As both trials demonstrated a positive change in the measured mediator, no further sensitivity analysis was conducted.

Three trials employed medication adherence scales: MMAS [44, 45]; MARS [43]. Data, as presented, permitted meta-analysis from two trials [43, 45] demonstrating a significant effect in favour of the intervention group (SMD 0.36 [0.07, 0.64], p = 0.01, $I^2$ 13%) and evidence of effect for the TDF domain of *Beliefs about Consequences*, drawn from one study [43] (Fig 3b).

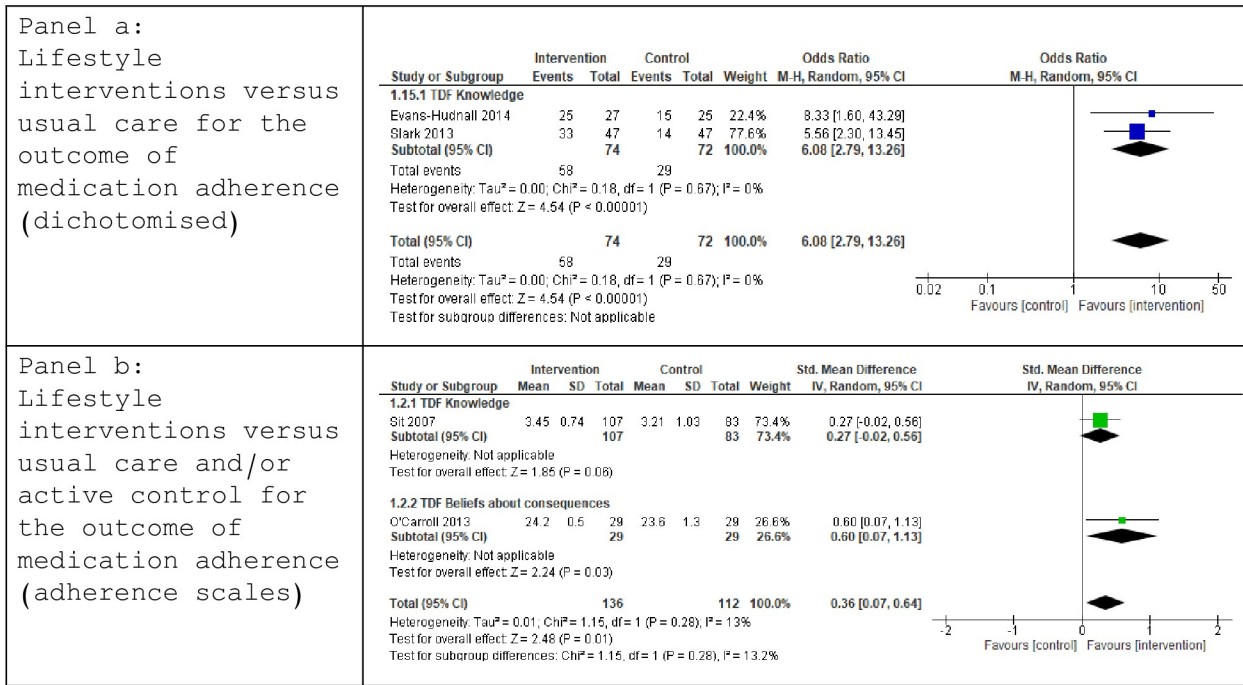

| Panel a: Lifestyle interventions versus usual care for the outcome of medication adherence (dichotomised) | |
| Panel b: Lifestyle interventions versus usual care and/or active control for the outcome of medication adherence (adherence scales) | |

**Fig 3. Forest plot: Lifestyle interventions versus usual care and/or active control for the outcome of medication adherence.**

GRADE criteria identified very low certainty of evidence (Table 2). No sensitivity analysis was conducted as both included trials demonstrated a positive change in their identified mediator.

**Diet.** Five trials reported healthy eating outcomes [34, 35, 45, 46, 48]. Data were pooled from three trials [34, 35, 48] with self-reported fruit and vegetable consumption outcomes and behaviour-change mediators mapped to TDF domains of *Knowledge* [34] and *Intentions* [35, 48]. Meta-analysis demonstrated no significant difference in fruit and vegetable consumption between groups (SMD -0.07 [-0.36, 0.21], p = 0.62, $I^2$ 0%) (S5 Fig). Sensitivity analysis, removing trials where no change in the mediator occurred [35, 48], left one remaining trial [34] that didn't alter the findings (Z = 0.04, p = 0.97).

Two trials reported outcomes addressing salt or salty food consumption [45, 46] and used behaviour-change mediators mapped to the TDF *Knowledge* domain. The units or mode of measurement employed did not allow data to be pooled. One of these trials [45] demonstrated a significant reduction in salted food consumption in the intervention group (p = 0.004).

**Physical activity participation.** Four trials reported physical activity participation outcomes [34, 35, 45, 46]. Data were pooled from two trials measuring self-reported physical activity levels [34, 35] and using behaviour-change mediators mapped to TDF domains of *Knowledge* [34] and *Intentions* [35]. Results demonstrated no significant effect in favour of the intervention group (SMD -0.07 [-0.46, 0.32], p = 0.72, $I^2$ = 67%), however heterogeneity was substantial (S6 Fig). Sensitivity analysis, removing one trial where the measured mediator didn't change [35], left one trial [34] that did not alter the findings (Z = 1.48, p = 0.14).

Data were pooled from two trials [45, 46] measuring self-reported achievement of physical activity targets and using behaviour-change mediators mapped to the TDF *Knowledge* domain. A significant effect in favour of the intervention was observed (OR 2.97 [1.73, 5.12], p< 0.0001, $I^2$ 0%)(Fig 4c). GRADE criteria identified Low certainty evidence (Table 2). No

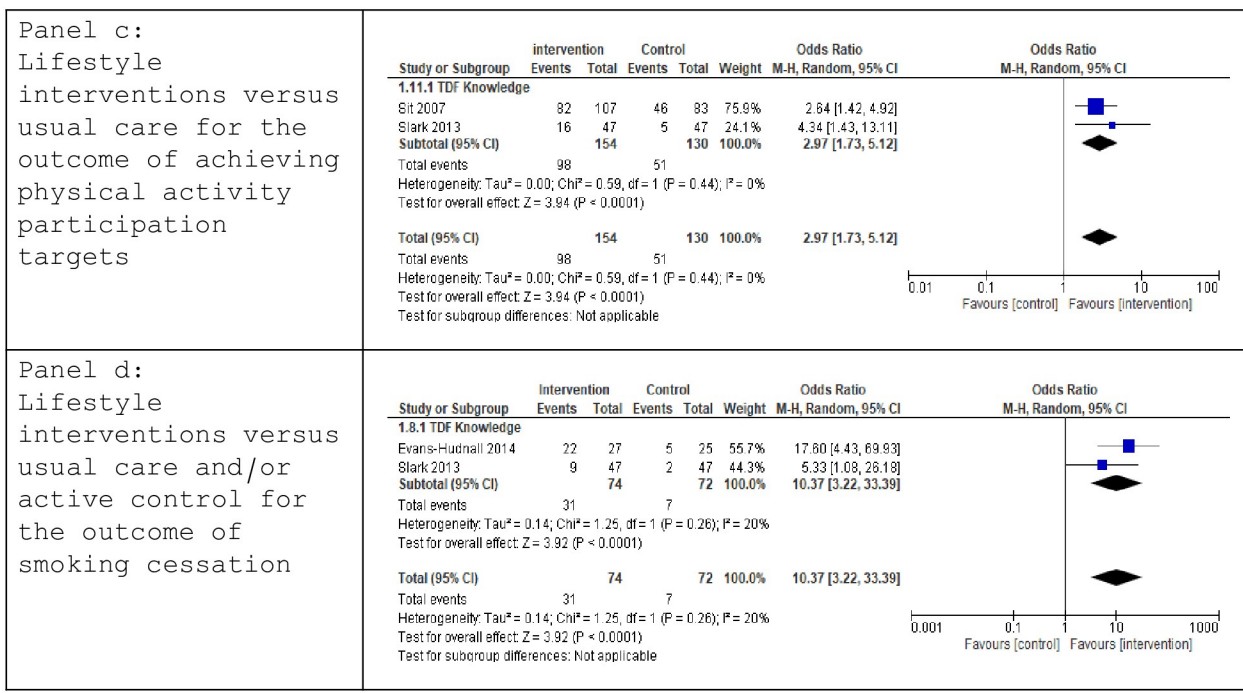

**Fig 4.** Forest Plot: Lifestyle interventions versus usual care for the outcome of achieving physical activity participation targets (panel c); Lifestyle interventions versus usual care and/or active control for the outcome of smoking cessation (panel d).

sensitivity analysis was conducted as both trials demonstrated positive changes in their measured mediator for behaviour-change.

**Smoking cessation.** Three trials addressed tobacco use post-stroke, employing behaviour-change mediators mapped to TDF domains of *Knowledge* [34, 46] and *Intention* [48]. Data were pooled for outcomes of smoking cessation rates post-intervention [34, 46]. Meta-analysis demonstrated a significantly higher likelihood of smoking cessation favouring the intervention group (OR 10.37 [3.22, 33.39], p < 0.0001, $I^2$ 20%) (Fig 4d). Low GRADE certainty evidence was identified (Table 2). Both trials demonstrated positive changes in the mediator measured, negating further sensitivity analysis.

**Alcohol consumption.** Three trials reported safe alcohol consumption outcomes using behaviour-change mediators mapped to TDF domains of *Knowledge* [34, 46] and *Intentions* [35]. The units or mode of measurement employed in reported outcomes (alcohol abstinence, units consumed per week, percentage of self-reported alcohol reduction) did not allow data to be pooled. However, two trials demonstrating positive change in the mediator of *Knowledge*, reported significant alcohol abstinence rates (OR = 1.48, 1.36–1.53, p = 0.31) [34] and alcohol consumption reduction (OR = 4.48 [1.16, 17.29], p = 0.03) [46] in favour of the intervention group. The remaining trial [35], where the mediator mapped to the *Intentions* domain, demonstrated no effect.

## Outcomes of emotional self-regulation

**Anxiety.** Four trials reported outcomes addressing anxiety self-regulation as a risk reducing behaviour, using mediators mapped to the TDF domains of *Knowledge* [33], *Beliefs about Capabilities* [38, 39] and *Emotions* [41]. Data, as presented in these trials, permitted meta-analysis from 3 trials [33, 38, 39] using the HADS-Anxiety sub-scale with negative results (MD

-0.68 [-1.63, 0.28], p = 0.26, I² = 0%) (S7 Fig). No trial demonstrated a positive change in their identified mediator, negating further sensitivity analysis.

**Depression.** Seven trials reported outcomes addressing self-regulation of low mood [33, 37–39, 41, 51, 52]. Data permitted meta-analysis from four trials using behaviour-change mediators mapped to TDF domains of *Beliefs about Capabilities* [38, 39] and *Emotions* [51, 52], demonstrating evidence of effect in favour of the intervention (SMD -0.70 [-1.28, -0.12], p = 0.02, I² = 81%), with high heterogeneity (Fig 5e). Evidence of effect in the TDF *Emotions* domain, drawn from 2 trials (SMD -0.99 [-1.97, -0.00], p = 0.05, I² = 91%) is also evident, however heterogeneity was considerable. Low GRADE certainty was applied to the main finding (Table 2). Sensitivity analysis, removing trials with no associated change in the mediator, left two remaining trials [51, 52] and did not alter the overall findings (Z = 1.96, p = 0.05, I² = 91%).

**Psychological distress.** Seven trials reported outcomes addressing self-regulation of psychosocial distress, using behaviour-change mediators mapped to TDF domains of *Beliefs about Capabilities* [39, 40, 42], *Beliefs about Consequences* [50], *Intentions* [35] and *Emotions* [37, 47]. Data presented permitted meta-analysis from four trials with no evidence of significant effect evident (SMD -0.12 [-0.28, 0.05], p = 0.18, I² = 0%) (S8 Fig). No trial demonstrated a positive change in the mediator measured, negating further sensitivity analysis.

## Discussion

Use of theory and theoretical constructs is recommended to enhance effectiveness of complex interventions [17, 55] and to allow greater understanding of behaviour-change processes [22]. This overview of reviews, providing new meta-analyses, summarises the quantity and quality of evidence supporting theoretically-grounded interventions for behavioural change and/or self-management of health behaviours in stroke secondary prevention. As multiple behaviour-change theories exist, the TDF proved a useful theoretical lens through which theory-associated mediators for behaviour-change post-stroke could be viewed. Formal mediation analysis (indirect and direct paths) was beyond the scope of this current work, rather the effect of proposed mediators associated with behaviour-change interventions was examined. In excluding RCTs unless a theoretical perspective and an identified and measured mediator for behaviour-change was provided, trials with inadequately described theoretical underpinnings were omitted. This addresses current uncertainty in published meta-analysis, where ineffectiveness or

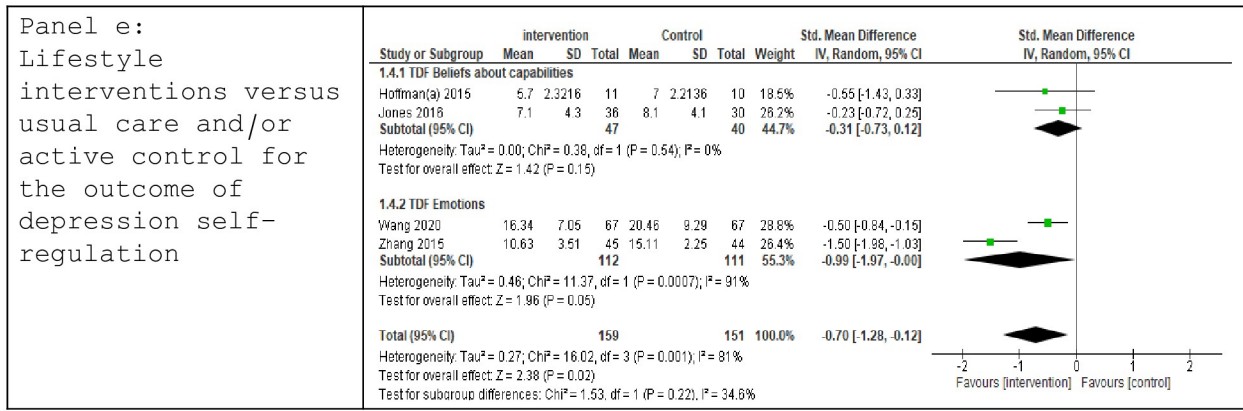

**Fig 5. Forest plot: Lifestyle interventions versus usual care and/or active control for the outcome of depression self-regulation.**

conflicting results may result from underuse of theory and/or a lack of understanding of determinants of health behaviours post-stroke.

Across the nineteen RCTs identified in this review, three TDF domains were associated with impactful health behaviour-change: *Knowledge* (increased stroke knowledge); *Beliefs about Consequences* (greater understanding of benefits/rationale for preventive actions); and *Emotions* (self-regulation of emotional responses). Indeterminate effects were found, with considerable heterogeneity, for two other TDF domains, of *Beliefs about Capabilities* (self-confidence, self-efficacy in ability to put knowledge to constructive use) and *Intentions* (conscious decision to perform a behaviour). The identification of only five TDF domains from a possible fourteen points to significant underutilisation and measurement of theory-associated mediators in current trials. As no trials addressing outcomes of cardiovascular morbidity, recurrent stroke, or other cardiovascular events in stroke secondary prevention met the inclusion criteria, evidence is still lacking for these primary outcomes. An important observation about the role of mediators in achieving health behaviour-changes post-stroke is evident across the meta-analyses reported. With the exception of the outcome of depression, all other secondary prevention outcomes demonstrating a positive effect (medication adherence, achievement of physical activity targets, smoking cessation), were drawn from trials where the mediator for change was positively influenced by the intervention. Similarly for negative meta-analyses related to fruit and vegetable consumption, anxiety and psychological distress outcomes, the proposed mediators for change did not improve in the included trials.

It is important to note commonalities and discrepancies that exist between the results of previously published SRs [11–15], a best evidence synthesis from these reviews [16] and the results presented here. High-level evidence of published meta-analyses addressing behavioural change and/or self-management interventions in stroke secondary prevention found only moderate overlap of primary studies between reviews, and identified the SRs as addressing different intervention types, broadly categorized as Multimodal; Behavioural change; Self-management and Psychological therapies [16]. This approach failed to delineate any theoretical overlap between the intervention types, or the specific mediators targeted in the interventions that were effective. The ability to replicate the intervention clinically and include the important theoretical domains by which to affect change remained challenging. This review now indicates that TDF domains of *Knowledge*, *Beliefs about Consequences*, and *Emotions* are important considerations for inclusion in secondary prevention interventions designed to achieve behaviour-change.

Current best-evidence identified in the TDF *Knowledge* domain is drawn from trials where mediators for behaviour-change included enhanced knowledge about the nature of stroke, risk factors, and consequences. Interventions provided interactive and tailored information on risk reducing behaviours; offered self-management information and support in goal setting to improve lifestyle habits; and provided information on self-monitoring in relation to maintaining behaviour-change. Four trials specified and measured stroke knowledge as the mediator for change associated with their intervention [33, 34, 45, 46]. Health behaviours that were successfully targeted through the mediator of stroke knowledge were medication adherence, physical activity participation, healthy eating, smoking and alcohol consumption. Published psychological determinants of medication adherence as a health behaviour, highlight the importance of knowledge for stroke survivors [56], noting greater knowledge is associated with better adherence to prescribed medication. Qualitative research further highlights that inadequate information about stroke and commonly prescribed preventive medications, pose significant patient-level barriers [57] and that passive written information provision was not a helpful means to improve adherence [58]. A recent scoping review identified a mismatch between guideline recommendations for behavioural counselling and that audited in clinical

practice, where the latter largely comprised information provision only [59]. Making and sustaining behaviour-change is difficult to achieve. Information provision alone has been shown to be ineffective in affecting the mediator of knowledge and thus positive behaviour-change [60]. Active rather than passive information provision, mirrored in the interventions of the included studies in this review, has been found to be effective [61]. This current overview now provides definitive evidence to guide clinical practice in stroke secondary prevention in the use of active mechanisms of information provision, anchored in the *Knowledge* domain of the TDF.

Addressing individuals' *Beliefs about Consequences* of stroke-related behaviours was identified in this overview as an important TDF domain that can influence adherence to preventive medications and target self-regulation of mood. This now constitutes an important domain to address in stroke secondary prevention interventions where adherence rates with prescribed medications is poor [62] and low mood is associated with higher stroke recurrence rates [63] and mortality [1]. Two trials identified in this overview sought to enhance individuals' beliefs and understanding of the benefits/rationale for preventive actions as their mediator for change [43, 50] but for specific outcomes (e.g. medication adherence), evidence for this TDF domain relies on single study data [43]. However, other research does support the relationship between perceived necessity for medications and adherence rates [56, 57].

In the TDF *Emotions* domain, three distinct modifiable mediators (coping competence, sense of coherence and trait mindfulness) were identified in the five included trials targeting emotional self-regulation to reduce post-stroke risk [37, 41, 47, 51, 52]. Of these, trait mindfulness, measured using MAAS, demonstrated positive change as the mediator to affect emotional self-regulation behaviour-change in two trials [51, 52]. Mindfulness, a state of awareness and attention to the present, has been shown by systematic review in stroke to derive benefits across psychological, physiological, and psychosocial outcomes including stroke secondary prevention targets of anxiety, depression and blood pressure [64]. Therapeutic benefits of mindfulness for individuals post-stroke, both psychological and physical, are supported by a recent scoping review. However, caveats about knowledge gaps relating to Mindfulness-based Cognitive Therapy (MBCT) and the lack of methodological robust studies were highlighted [65].

The *Beliefs about Capabilities* TDF domain, with self-efficacy as the mediator for behaviour-change, was identified in five trials [38–40, 42, 44] and the *Intentions* TDF domain, using stages of change as the mediator, was identified in three trials [35, 36, 48]. Whilst self-efficacy was the mediator most frequently employed in primary studies interrogated, no trial employing self-efficacy as the mediator, was associated with health behaviour-changes post-stroke. This finding was unexpected. Self-efficacy is a well-established predictor of behaviour-change relating to physical activity participation for example [10], with a recent prospective cohort study identifying self-efficacy as the strongest contributing factor to stroke/TIA survivors' intentions to change health behaviours [66]. Perceived self-efficacy is the main causal determinant cited in Social Cognitive Theory [67], is the proposed mediator for change in The Stanford University Chronic Disease Self-Management programme [68] and the Bridges Stroke Self-Management Programme [69]. Whilst individuals' self-efficacy may have responded positively to the interventions included in the current review, results did not translate to risk-reducing behaviour-change. Likewise, included trials [35, 48] with a mediator anchored in the TDF *Intentions* domain reported conflicting results between mediators and outcomes. This finding may relate, in part, to poor alignment between theory as a set of concepts and propositions that explain behaviour, and the subsequent use of this theory to then identify and target the determinants of behaviour to affect behaviour-change [27]. This was not clearly articulated in the majority of included trials which also failed to measure all mediators cited in their theory

or to link the intervention techniques employed to these constructs, as best practice recommends.

Risk of stroke, stroke recurrence and both short and long-term outcomes disproportionally affect low and middle-income countries [2] where healthcare systems are primarily focused on acute care [70] and inconsistencies in ongoing care and service delivery exist [71]. In addition, deprived or disadvantaged populations within high-income countries where modifiable risk factors are prevalent, experience disparities [72]. Factors such as income, education, social support or isolation, ethnicity, and environmental conditions have been identified as confounding variables that need to be addressed in intervention studies [73, 74]. One study included in the current review showed a positive change in tobacco and alcohol use with African American and Hispanic participants of low socioeconomic status [34]. However, despite the disproportionate burden of stroke on low and middle-income countries, the majority of the papers reviewed originated in high-income countries, limiting the generalisability of the findings. Whilst beyond the scope of this overview, we recognise the need for a better understanding and management of the gap observed between high- and low-income nations when designing and delivering effective stroke secondary prevention interventions.

Overall, results from this overview, which interrogated primary study-level data, need to be interpreted carefully. Despite a rigorous mediator identification process and mapping to an established framework (TDF), an element of subjectivity in this process remains. This is largely dependent on interpreting the definitions and descriptions provided in the primary studies and the mediator measured. Because strict criteria were applied for trial inclusion, evidence identified was often downgraded to low or very low due to the limited number of studies, small sample sizes, and risks of bias. Nevertheless, having GRADE for all included effective interventions is a strength of this overview. The limited number of trials available results, in the main, from unclear reporting of behaviour-change interventions in the RCTs screened, a recurrent issue in the extant literature. The growing number of complex and multicomponent interventions focusing on stroke secondary prevention [75], when recently scrutinised using the updated MRC Framework [20], identified only a small proportion providing a theoretical underpinning, and none reporting their proposed mediators of action or specifying behaviour-change techniques [75]. Studies included in this overview presented wide variations in intervention content and delivery, timing after stroke, duration, location, and outcomes targeted, reflecting many of the challenges inherent in developing and reporting complex interventions in stroke secondary prevention [76]. The small number of trials identified, despite the large volume of SRs, highlights a lack of standardisation in how behaviour-change and self-management interventions are reported, particularly in relation to describing their rationale and theoretical perspectives (points 1 and 2 of the TIDieR checklist [23]).

Future studies should endeavour to describe their behaviour-change interventions in a structured way. This should lay out the use of theory, the theory-related mediator/s targeted to affect behaviour-change, alongside the matched behaviour-change techniques employed. To this end, using TDF domains to select evidence-based behaviour-change techniques (BCT) as the active ingredients of an intervention [77] has evolved with an extensive taxonomy of BCTs available to draw from [78], and recent advances in behaviour-change intervention ontologies to aid the development of effective interventions [79]. Future trials in behaviour-change for stroke secondary prevention, in addition to attending to how the intervention is developed and described, need to include outcomes that measure longer-term cardiovascular outcomes as well as behaviour-change, as these features were markedly absent in the reviewed literature.

## Conclusion

This overview highlights the current lack of theory-based research and limited use of behaviour-change mediators within stroke secondary prevention trials. The findings identify the theoretical domains of *Knowledge*, *Beliefs about Consequences*, and *Emotions* as having utility in affecting positive changes in risk reducing health behaviours post-stroke to actively increase knowledge and awareness of all aspects of stroke and risk-reducing behaviours; address health beliefs and correct any misperceptions; and support emotional self-regulation. Taken together with the findings from a best-evidence synthesis previously published [16], the active components of successful strategies in theoretically-based behaviour-change and self-management interventions are better explained. Future research should, at a minimum, include these constructs known to be effective, in well conceptualised RCTs, with adequate follow-up time, that include primary outcomes addressing cardiovascular risk in addition to health behaviours.

## Supporting information

**S1 Table. Theoretical Domains Framework.** ✓ denotes domains identified in primary studies included in current study.
(DOCX)

**S2 Table. Preferred reporting guideline for overviews of reviews of healthcare interventions (PRIOR) checklist.**
(DOCX)

**S3 Table. Template for intervention description and replication (TIDieR) checklist.**
(DOCX)

**S4 Table. Preferred reporting items for systematic reviews and meta-analyses (PRISMA) checklist.**
(DOCX)

**S1 File. Search strategy.**
(DOCX)

**S1 Fig. Model for person-centred stroke secondary prevention behavioural change and self-management.**
(TIF)

**S2 Fig. Summary of risk of bias.**
(TIF)

**S3 Fig. Risk of bias graph.** Review authors' judgements about each risk of bias item presented as percentages across all included studies.
(TIF)

**S4 Fig. Forest plot: Lifestyle interventions versus usual care for the outcome of mortality.**
(TIF)

**S5 Fig. Forest plot: Lifestyle interventions versus usual care for the outcome of fruit & vegetable consumption.**
(TIF)

**S6 Fig. Forest plot: Lifestyle interventions versus usual care for the outcome of physical activity participation levels.**
(TIF)

**S7 Fig. Forest plot: Lifestyle interventions versus usual care for the outcome of anxiety self-regulation.**
(TIF)

**S8 Fig. Forest plot: Lifestyle interventions versus usual care and/or active control for the outcome of psychological distress.**
(TIF)

## Acknowledgments

We acknowledge foundational work by the INSsPiRE network (International Network of Stroke Secondary Prevention Researchers) in identifying and conceptualizing the need for this overview of reviews.

## Author Contributions

**Conceptualization:** Maggie Lawrence, Olive Lennon.

**Data curation:** Patricia Hall, Olive Lennon.

**Formal analysis:** Patricia Hall.

**Funding acquisition:** Olive Lennon.

**Investigation:** Patricia Hall.

**Methodology:** Patricia Hall, James Matthews, Olive Lennon.

**Project administration:** Patricia Hall.

**Resources:** Olive Lennon.

**Supervision:** Maggie Lawrence, Thilo Kroll, Olive Lennon.

**Validation:** James Matthews, Olive Lennon.

**Writing – original draft:** Patricia Hall.

**Writing – review & editing:** Patricia Hall, Maggie Lawrence, Thilo Kroll, Catherine Blake, James Matthews, Olive Lennon.

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
