## [Decision Letter · Decision Letter 0]

23 Feb 2024

PONE-D-23-30783Reducing risk behaviours after stroke: an overview of reviews interrogating primary study data using the Theoretical Domains FrameworkPLOS ONE

Dear Dr. Hall,

Thank you for submitting your manuscript to PLOS ONE. After careful consideration, we feel that it has merit but does not fully meet PLOS ONE’s publication criteria as it currently stands. Therefore, we invite you to submit a revised version of the manuscript that addresses the points raised during the review process.

We look forward to receiving your revised manuscript.

Kind regards,

Tinashe Mudzviti, MPhil(MD)

Academic Editor

PLOS ONE

Journal Requirements:

Reviewers' comments:

Reviewer's Responses to Questions

**Comments to the Author**

1. Is the manuscript technically sound, and do the data support the conclusions?

Reviewer #1: Yes

Reviewer #2: Yes

2. Has the statistical analysis been performed appropriately and rigorously? 

Reviewer #1: Yes

Reviewer #2: No

3. Have the authors made all data underlying the findings in their manuscript fully available?

Reviewer #1: Yes

Reviewer #2: Yes

4. Is the manuscript presented in an intelligible fashion and written in standard English?

Reviewer #1: Yes

Reviewer #2: Yes

5. Review Comments to the Author

Reviewer #1: The manuscript presented for review is articulate and demonstrates adequate methodological rigor.

However, I would like to offer a few constructive comments to enhance its academic merit:

1. SMD definition: You should specify the definition of Standardized Mean Difference (SMD) utilized in your analysis. Given that you have employed RevMan, unless there have been alterations to the default settings, it should be Hedges's g. Please verify this is the case and appropriately detail it in the methods section.

2. Random Effects Model: In your study, you have implemented a random effects model, employing the Mantel-Haenszel method for dichotomous variables and the Inverse Variance method for continuous variables. This approach is appropriate for the context of your analysis. Nonetheless, it is essential to explicitly state this choice and adequately elucidate its rationale in the methods section.

3. Tabular Representation of Theoretical Framework Domains (TFD):I would suggest to include a table that succinctly encapsulates the TFD. This table should feature, in one column, each domain, and in the adjacent column, a description of these domains as they pertain to your study's context. Although a similar approach is partially employed in the main text (lines 387-389), it would be beneficial to extend this to encompass domains not found in your study. This will not only provide clarity but also serve as a valuable guide for future research endeavors.

4. Enhanced Detail in Table 1 description of Interventions: The inclusion of an additional column in Table 1, offering a more comprehensive description of each intervention, is advisable for clarity. Presently, the connection between certain components and their corresponding interventions is not immediately clear. Indeed, while certain intervention names are self-explanatory, others are not as immediately apparent in their relationship. For example, the association of the emotional component in Hjelle et al., 2019, remains unclear in the table when only the name of the intervention is provided. To remedy this, I would suggest including a brief description for each intervention in a new column. This addition will clarify the relevance and application of each intervention within the context of your study, ensuring a more comprehensive understanding for the reader and elucidating framework associations.

5. Resolution of Images: The images included in the manuscript in my possess are of low resolution, which detracts from the overall quality of the presentation.

6. High heterogeneity meta-analysis should be reported anyway: Although some meta-analyses may lack validity due to being characterized by high heterogeneity, as reported, it would be interesting and transparent to include them, possibly in the supplementary materials. Moreover, you have indeed reported results with high heterogeneity in lines 312 and 354-359, but these have not been identified as such.

Reviewer #2: 

While the manuscript is important to further advance our understanding on stroke, there are several things that the authors should address before it can be published. For my specific comments, see below.

Introduction:

The authors did not explain in detail what is TDF. Why is it important?

Methodology:

The authors did not state what is exclusion criteria for this review. They also did not give justification why they only used three search engines to search the articles. Why did they leave out other search engines like WoS and SCOPUS that might yield more articles? How sure are the authors that comprehensive search has been conducted to make sure no article is missing from this analysis?

Discussion:

Please discuss why most studies included here come from high income nations? Is there any gap observed between high income and low income nations?

Minor correction: Figure presented in manuscript is not clear and hard to read.

6. PLOS authors have the option to publish the peer review history of their article (what does this mean?). If published, this will include your full peer review and any attached files.

Reviewer #1: **Yes: **Pierfelice Cutrufelli

Reviewer #2: No

---

## [Author Response · Author response to Decision Letter 0]

16 Mar 2024

Thank you for your positive comments and detailed feedback on our paper. Below is a point-by-point response to each point raised by the academic editor and reviewer(s).

Reviewer #1 Points 1 - 6

Reviewer #1 comments to authors: The manuscript presented for review is articulate and demonstrates adequate methodological rigor. However, I would like to offer a few constructive comments to enhance its academic merit.

Response: We greatly appreciate the positive comments provided in your detailed feedback and thank you for your review.

Point 1

1. SMD definition: You should specify the definition of Standardized Mean Difference (SMD) utilized in your analysis. Given that you have employed RevMan, unless there have been alterations to the default settings, it should be Hedges's g. Please verify this is the case and appropriately detail it in the methods section.

Response: Thank you, the SMD used in the analysis is Hedges’ g and we have detailed this in the data synthesis of the methods section which now reads:

Meta-analyses, conducted using Review Manager 5 (RevMan5)[29], grouped data by TDF domains and secondary prevention outcomes, where data presented permitted. For continuous data, where different scales assessed the same outcome, standardized mean differences (SMD) with 95% confidence intervals (CI) were calculated. SMD is used as a summary statistic to measure effect size that quantifies differences in standard deviations between two groups. The Hedges’ g version of SMD conducted here in RevMan5 is the preferred statistic when sample sizes are unequal and/or are small (< 20), as the case in the current study, as it takes each sample size into consideration when calculating the overall effect size. 

Point 2

2. Random Effects Model: In your study, you have implemented a random effects model, employing the Mantel-Haenszel method for dichotomous variables and the Inverse Variance method for continuous variables. This approach is appropriate for the context of your analysis. Nonetheless, it is essential to explicitly state this choice and adequately elucidate its rationale in the methods section.

Response: Thank you we have amended the methods section to provide the necessary clarity. This section now reads in the manuscript:

For continuous data where different scales assessed the same outcome, standardized mean differences (SMD) with 95% confidence intervals (CI) were calculated. SMD is used as a summary statistic to measure effect size that quantifies differences in standard deviations between two groups. The Hedges’ g version of SMD conducted here in RevMan5 is the preferred statistic when sample sizes are unequal and/or are small (< 20), as the case in the current study, as it takes each sample size into consideration when calculating the overall effect size. The inverse variance method was used as it is especially suitable when using SMD to minimise uncertainty of the overall effect size[30]. For dichotomous variables, odds ratios (OR) with 95% CIs were employed using the Mantel-Haenszel method. Random effects models were applied to provide a more conservative estimate of overall effect size as statistical heterogeneity was assumed[30]. The I2 statistic measured heterogeneity; >50% was considered substantial.

Point 3

3. Tabular Representation of Theoretical Framework Domains (TFD): I would suggest to include a table that succinctly encapsulates the TFD. This table should feature, in one column, each domain, and in the adjacent column, a description of these domains as they pertain to your study's context. Although a similar approach is partially employed in the main text (lines 387-389), it would be beneficial to extend this to encompass domains not found in your study. This will not only provide clarity but also serve as a valuable guide for future research endeavors.

Response: Thank you for this suggestion. We have now included a TDF table in the supplemental file and referenced this in the introduction which now reads:

Comprising 87 component parts across fourteen overarching domains of Knowledge, Skills, Social/Professional Role and Identity, Beliefs about Capabilities, Optimism, Beliefs about Consequences, Reinforcement, Intentions, Goals, Memory, Attention and Decision Processes, Environmental Context and Resources, Social Influences, Emotions, and Behavioural Regulation[22], the TDF provides comprehensive coverage of the possible mediators influencing behaviour-change. Tabular representation of the TDF describes these domains as they pertain to stroke secondary prevention in the current study (supplemental file, S1 Table).

Point 4

4. Enhanced Detail in Table 1 description of Interventions: The inclusion of an additional column in Table 1, offering a more comprehensive description of each intervention, is advisable for clarity. Presently, the connection between certain components and their corresponding interventions is not immediately clear. Indeed, while certain intervention names are self-explanatory, others are not as immediately apparent in their relationship. For example, the association of the emotional component in Hjelle et al., 2019, remains unclear in the table when only the name of the intervention is provided. To remedy this, I would suggest including a brief description for each intervention in a new column. This addition will clarify the relevance and application of each intervention within the context of your study, ensuring a more comprehensive understanding for the reader and elucidating framework associations.

Response: Thank you for this suggestion. An additional column has been inserted in the characteristics table to provide a brief description of each intervention to aid clarity. The characteristics table headings now read as below with additional column (full table not included here but title row copied below for your convenience).

Study Participants Intervention theoretical perspectives Brief description Mediator TDF Domain Intervention Time to follow-up Outcomes measured Key findings

Point 5

5. Resolution of Images: The images included in the manuscript in my possess are of low resolution, which detracts from the overall quality of the presentation.

Response: Thank you. All images will now be uploaded to the Preflight Analysis and Conversion Engine (PACE) as now advised to ensure quality of figures.

Point 6

6. High heterogeneity meta-analysis should be reported anyway: Although some meta-analyses may lack validity due to being characterized by high heterogeneity, as reported, it would be interesting and transparent to include them, possibly in the supplementary materials. Moreover, you have indeed reported results with high heterogeneity in lines 312 and 354-359, but these have not been identified as such.

Response: Thank you. We have included all meta-analyses conducted either directly in the manuscript or in supplemental material irrespective of heterogeneity identified. Where some confusion may have arisen is where studies employed different outcome measures that did not allow meta-analyses to be conducted for example – salt and salty food consumption; alcohol abstinence, units consumed per week, percentage of self-reported alcohol reduction. We previously reported this as “heterogeneity in reported outcomes prohibited meta-analysis”. 

We have reworded this in the manuscript in the Diet and the Alcohol consumption section of the Results which now reads: The units or mode of measurement employed did not allow data to be pooled.

We have clarified in the methods section the level of heterogeneity considered which now reads:

The I2 statistic measured heterogeneity; >50% was considered substantial[30].

In addition, in the results section where we have reported results with high heterogeneity, we have now identified these as so. The manuscript now reads:

Physical activity participation

Results demonstrated no significant effect in favour of the intervention group (SMD -0.07 [-0.46, 0.32], p = 0.72, I2 = 67%), however heterogeneity was substantial (S6 Fig).

Depression

Data permitted meta-analysis from four trials using behaviour-change mediators mapped to TDF domains of Beliefs about Capabilities [37, 38] and Emotions[50, 51], demonstrating evidence of effect in favour of the intervention (SMD -0.70 [-1.28, -0.12], p = 0.02, I2 = 81%), with high heterogeneity (Fig 5e). Evidence of effect in the TDF Emotions domain, drawn from 2 trials (SMD -0.99 [-1.97, -0.00], p=0.05, I2 = 91%) is also evident, however heterogeneity was considerable.

Reviewer #2 Comments 1 - 4

Reviewer #2 comments to authors: While the manuscript is important to further advance our understanding on stroke, there are several things that the authors should address before it can be published. For my specific comments, see below.

Response: We very much appreciate your positive comments and thank you for your review. Outlined below are point-to-point responses to each specific comment.

Comment 1 Introduction:

The authors did not explain in detail what is TDF. Why is it important?

Response: Thank you. We have amended the introduction to include further details to explain the TDF and it’s importance, and have included a TDF table in a supplemental file to provide greater clarity, referenced in the introduction which now reads:

Explicit use of theory in the design and evaluation of interventions in stroke secondary prevention presents this opportunity to understand why interventions work, for whom, and in what context[18]. Selecting one or more theories as the basis for intervention development can prove challenging, partly due to often overlapping theoretical constructs[21]. The Theoretical Domains Framework (TDF) is a comprehensive theory-informed approach to understanding the determinants of behaviour change and the factors influencing intervention development. The TDF was developed to allow theories and their constructs to be synthesised into groupings to make behaviour-change theories more accessible in intervention design and analysis[21, 22]. This is important as it provides a systematic and rigorous framework for understanding behaviour change in multiple populations and settings. Comprising 87 component parts across fourteen overarching domains of Knowledge, Skills, Social/Professional Role and Identity, Beliefs about Capabilities, Optimism, Beliefs about Consequences, Reinforcement, Intentions, Goals, Memory, Attention and Decision Processes, Environmental Context and Resources, Social Influences, Emotions, and Behavioural Regulation[22], the TDF provides comprehensive coverage of the possible mediators influencing behaviour-change. Tabular representation of the TDF describes these domains as they pertain to stroke secondary prevention in the current study (supplemental file, S1 Table).

Comment 2 Methodology:

The authors did not state what is exclusion criteria for this review. 

Response: Thank you for your feedback. Apologies, we stated the exclusion criteria in the screening and selection section and have amended the manuscript by moving this to explicitly state it following the inclusion criteria. The manuscript now reads:

Inclusion/Exclusion criteria

SRs of randomized control trials (RCTs) or cluster RCTs (CRCT) testing interventions for behaviour-change and/or self-management of risk in stroke secondary prevention were first identified. Primary studies included in these reviews were then considered where the following were detailed: 

• Adult population comprising stroke/TIA 

• Intervention/s targeting stroke risk reduction at an individual or population level 

• Intervention/s identifying a theoretical perspective and measuring a stated mediator for behaviour-change that mapped to the TDF

• Comparators of usual care, placebo, sham, or other intervention

• Outcomes recorded that addressed mortality, recurrent stroke or other cardiovascular events, or secondary outcomes addressing any one or combination of the following health behaviours – secondary prevention medication adherence, healthy diet, physical activity participation, smoking cessation, safe alcohol consumption and emotional self-regulation. 

Exclusion criteria applied:

• Interventions designed to alter care process or health professionals’ education/practice.

• Interventions not targeting behaviour-change in stroke secondary prevention. 

• Telehealth interventions 

• Interventions targeting family/partner dyads, unless behaviour-change in the person with stroke was specifically targeted and extractable.

They also did not give justification why they only used three search engines to search the articles. Why did they leave out other search engines like WoS and SCOPUS that might yield more articles? How sure are the authors that comprehensive search has been conducted to make sure no article is missing from this analysis?

Response: Thank you. We worked closely with the information scientist (librarian) on the search strings using controlled vocabulary and free text terms relating to stroke and secondary prevention/lifestyle risk behaviour which we developed and adapted for the included databases – Epistemonikos, Cochrane Library of Systematic Reviews, Medline, Embase. These libraries were chosen as they represent two systematic review databases and two of the largest health focused databases. For the databases not specific to systematic reviews (Medline, Embase), a third methodological search string for systematic reviews was included. Based on our experience and previous searches we were confident these databases would contain the reviews we were looking for. As an overview of systematic reviews only primary studies included in these reviews were included therefore it is possible that more recent RCTs are not included.

To clarify this in the Search strategy section the manuscript now reads: 

Using a comprehensive search strategy compiled in conjunction with a liaison librarian, electronic databases MEDLINE, Embase, Epistemonikos and Cochrane Library of Systematic Reviews were systematically searched from inception to March 2023 with no limitations applied. For databases not specific to systematic reviews (Medline, Embase), a third methodological search string for systematic reviews was included. These two databases were chosen as they are two of the largest health focussed databases and we were confident, based on our experience and previous searches that they would contain the reviews we were looking for. In addition, reference lists of included SRs were checked. It is possible that more recent RCTs, not yet reviewed in SRs are not included. The full search strategy which targeted published systematic reviews is provided (S1 File). 

Comment 3 Discussion:

Please discuss why most studies included here come from high income nations? Is there any gap observed between high income and low income nations?

Response: Thank you for your comments and support for this paper. We welcome your observations related to the significant gap in the global burden of stroke between high-income and low-income nations. Our review purposely focused on the available literature from identified systematic reviews, of theoretically-grounded interventions that applied mediators for behaviour-change. Despite the disproportionate burden of stroke on low and middle income countries, the majority of the papers originated in high-income countries. 

We have clarified this in the description of the included RCTs in the results section which now reads: 

Table 1 summarises the characteristics of all nineteen included RCTs. The trials were conducted across four geographical locations – Australia[32, 37, 39], North America[33, 35, 47], Asia[43, 44, 50, 51] and Europe[34, 36, 38, 40-42, 45, 46, 49]. When broken down by country and world bank classification all but two studies from an upper-middle-income economy (China)[51, 52] originated in high-income nations[54]. 

Response: thank you for your feedback. We now address this observed gap in the discussion section which reads in the manuscript: 

Risk of stroke, stroke recurrence and both short and long-term outcomes disproportionally affect low and middle-income countries[2] where healthcare systems are primarily focused on acute

---

## [Editor Report · Decision Letter 1]

3 Apr 2024

Reducing risk behaviours after stroke: an overview of reviews interrogating primary study data using the Theoretical Domains Framework

PONE-D-23-30783R1

Dear Dr. Hall,

We’re pleased to inform you that your manuscript has been judged scientifically suitable for publication and will be formally accepted for publication once it meets all outstanding technical requirements.

Kind regards,

Tinashe Mudzviti, MPhil(MD)

Academic Editor

PLOS ONE